

# Genetic diversity of enterotoxigenic *Bacillus cereus* strains in coriander in southwestern Mexico

Daniel Alexander Castulo-Arcos[1], Roberto Adame-Gómez[1], Natividad Castro-Alarcón[2], Aketzalli Galán-Luciano[1], María Cristina Santiago Dionisio[3], Marco A. Leyva-Vázquez[4], Jose-Humberto Perez-Olais[5], Jeiry Toribio-Jiménez[6] and Arturo Ramirez-Peralta[1]

[1] Laboratorio de Investigación en Patometabolismo Microbiano/Facultad de Ciencias Químico Biológicas, Universidad Autonoma de Guerrero, Chilpancingo, Guerrero, Mexico
[2] Laboratorio de Investigación en Microbiología/Facultad de Ciencias Químico Biológicas, Universidad Autonoma de Guerrero, Chilpancingo, Guerrero, México
[3] Laboratorio de Investigación en Análisis Microbiológicos/Facultad de Ciencias Químico Biológicas, Universidad Autonoma de Guerrero, Chilpancingo, Guerrero, México
[4] Laboratorio de Investigación en Biomedicina Molecular/Facultad de Ciencias Químico Biológicas, Universidad Autonoma de Guerrero, Chilpancingo, Guerrero, México
[5] Laboratorio de Biología Celular/Unidad Cuajimalpa, Universidad Autonoma Metropolitana, Ciudad de México, Ciudad de México, México
[6] Laboratorio de Investigacion en Microbiologia Molecular y Biotecnologia Ambiental/Facultad de Ciencias Químico Biológicas, Universidad Autonoma de Guerrero, Chilpancingo, Guerrero, Mexico

Corresponding author
Arturo Ramirez-Peralta,
ramirezperaltauagro@gmail.com

## ABSTRACT

**Background:** Coriander, like other leafy green vegetables, is available all year round and is commonly consumed raw in Mexico as in other countries in the preparation of street or homemade food. *Bacillus cereus* (*B. cereus*) is a microorganism that can reach coriander because it is usually found in the soil and in some regions the vegetables are irrigated with polluted water. Therefore, the aim of this study was to determinate the presence of *B. cereus* in coriander used for human consumption in southwestern Mexico and determine the toxigenic profile, biofilm production, genes associated with the production of biofilms, sporulation rates, enzymatic profile, psychotropic properties, and genetic diversity of *B. cereus*.

**Methods:** Fresh coriander samples were collected from several vegetable retailers in different markets, microbiological analysis was performed. Molecular identification, genes related to the production of biofilm, and toxin gene profiling of *B. cereus* isolates were determined by PCR. The biofilm formation was measured by performing a crystal violet assay. The genetic diversity of *B. cereus* strains was determined by PCR of repetitive elements using oligonucleotide (GTG) 5.

**Results:** We found a frequency of *B. cereus* in vegetables was 20% (13/65). In this study, no strains with genes for the HBL toxin were found. In the case of genes related to biofilms, the frequency was low for *sipW* [5.8%, (1/17)] and *tasA* [11.7%, (2/17)]. *B. cereus* strains produce a low amount of biofilm with sporulation rates around 80%. As for genetic diversity, we observed that strains isolated from the same market, but different vegetable retailers are grouped into clusters. In the coriander marketed in southwestern Mexico, were found *B. cereus* strains with genes associated with the

production of diarrheal toxins. Together, these results show actual information about the state of art of *B. cereus* strains circulating in the southwestern of Mexico.

# INTRODUCTION

Coriander is one of the most used species in gastronomy (*Laribi et al., 2015*), being Mexico one of the leading producers of this vegetable (*FAO, 2017*). By 2017, Mexico exported 64,000 tons of coriander, bound for five nations, highlighting the significant volume contributing to international markets, particularly the United States, which acquired 98.1% of that year's production (*SIAP, 2018*). Like other leafy green vegetables, coriander is available all year round and is commonly consumed raw in Mexico as in other countries (*Campbell et al., 2001*; *Gómez-Aldapa et al., 2016*). It has been associated with food poisoning by various microorganisms such as *Escherichia coli* O157 (*Whittaker et al., 2009*), *Salmonella* (*Campbell et al., 2001*), and *Cyclospora cayetanensis* (*C. cayetanensis*) (*Abanyie et al., 2015*). The study of the coriander microbiome has shown the presence of other microorganisms, including the order *Bacillales* (*Jarvis et al., 2015*). Within the *Bacillales* group is *Bacillus cereus* (*B. cereus*), a microorganism that can reach coriander because it is usually found in the soil. Furthermore, the formation of spores is considered an important mechanism of contamination and environmental resistance (*Vilain et al., 2009*).

 *B. cereus* share a high phenotypic, and genotypic similarity with other *Bacillus* species, which is why they have been included in a group called the *B. cereus group* or *B. cereus* sensu *lato* (*Helgason et al., 2000*). The *Bacillus cereus* sensu *lato* significantly impacts human health, agriculture, and food processing (*Rasko et al., 2005*). Various *Bacillus* strains have been used as biocontrol for phytopathogens in agriculture, including *Xanthomonas citri subsp. citri, Erysiphe heraclei, Macrophomina phaseolina, Fusarium oxysporum f. sp. Fragariae, Sclerotinia sclerotiorum*, tobacco streak virus, and tomato spot virus (*Wang et al., 2022*; *Hong et al., 2022*; *Saravanan et al., 2021*; *Ahmed et al., 2021*). Also, has been described the bioremediation activity of some strains against heavy metals and pyrethroid-type pesticides such as cypermethrin and fenpropathrin (*Bhatt et al., 2021*, *2020*; *Xiao et al., 2015*; *Chen et al., 2012*, *2014*; *Arora, 2020*). Even with the multiple applications of *Bacillus* strains, some strains of *B. cereus* can cause food poisoning due to the consumption of different products contaminated including vegetables, have also been described (*Yu et al., 2019*; *Park et al., 2018*).

 *B. cereus* is a foodborne pathogen that can cause two of clinical symptoms: emetic and diarrheal syndrome (*Stenfors Arnesen, Fagerlund & Granum, 2008*). The first is associated with the production of cereulide toxin, encoded by the *ces* operon (*Ehling-Schulz, Fricker & Scherer, 2004*). The second is caused by the combination of one or three enterotoxins described in *B. cereus*: non-hemolytic toxin (*NHE*), hemolytic toxin (HBL), and cytotoxin K (*CYTK*) (*Beecher et al., 2000*; *Lund, De Buyser & Granum, 2000*; *Lund & Granum, 1996*).

*B. cereus* has been previously reported in vegetables such as broccoli, carrots, lettuce, coriander (*Flores-Urbán et al., 2014*), spinach (*Tango et al., 2014*), and in fresh salads (*Becker et al., 2019*). The production of biofilms on the surface of lettuce has even been reported as a possible *B. cereus* persistence mechanism in the product (*Elhariry, 2011*). *B. cereus* in vegetables is relevant because the promotion and consumption of these products have increased due to their nutritional characteristics (*World Health Organization, 2003*; *Su & Arab, 2006*). However, most of these are consumed raw or minimally cooked, being considered the main route of exposure to this microorganism (*EFSA Panel on Biological Hazards (BIOHAZ), 2016*). Therefore, it is crucial to evaluate the toxigenic profile of the *B. cereus* strains from vegetables due to the potential described above and their differentiation from *B. thuringiensis*, which could be applied as a biopesticide and is not associated with food poisoning (*EFSA Panel on Biological Hazards (BIOHAZ), 2016*). Also, it is important to highlight the characteristics of the strains that allow them to be found in the final product, including the psychrophilic capacity, the production of biofilms, and spores. Finally, the information on *B. cereus* in Mexico in vegetables is scarce, so updating this information is necessary so that it can be included if needed, in current health legislation. Therefore, this study aims to determine the toxigenic profile, biofilm production, genes associated with the production of biofilms, sporulation rates, enzymatic profile, psychotropic properties and, genetic diversity of *B. cereus* strains from coriander in southwestern Mexico.

## MATERIALS AND METHODS

### Coriander samples

In this study, a total of 65 fresh coriander samples were included, these were purchased each week from 10 vegetable retailers from two municipal markets for 6 weeks and five samples of coriander purchased in the city's main supermarkets were included.
The coriander's samples were transported to the laboratory in sterile bags at room temperature. Once in the laboratory, the samples were analyzed without exceeding 1 h.

### Microbiological analysis

The microbiological analysis of coriander was only carried out with the leaves and not with the stems. Briefly, 25 g of coriander leaves were homogenized with 225 mL of 0.8% NaCl solution for 1 min. Once the samples were homogenized, additional 10-fold dilutions were made. The search for *B. cereus* was performed on Mannitol Egg Yolk Polymyxin (MYP) agar (Bioxon, Mexico). Later, 0.1 mL of the factorial dilutions were streaked on MYP agar and incubated for 24 h at 30 °C. We consider suspicious colonies of *B. cereus* those that show pink coloration with a halo of precipitation. The strains were confirmed as *B. cereus* by hemolytic capacity on 5% sheep blood agar (*ISO, 2004*).

### Molecular identification and toxin gene profiling of *B. cereus* isolates

From bacterial cultures, a thermal shock was performed to obtain the chromosomal DNA. In brief, cells from one colony were suspended in sterile water, heated at 95 °C for 3 min,

**Table 1 Polymerase chain reaction cycling conditions and primer sequences.**

| Gene | Primer sequences | PCR cycling conditions | Reference |
|---|---|---|---|
| *gyrB* | F-GCC CTG GTA TGT ATA TTG GAT CTA C R-GGT CAT AAT AAC TTC TAC AGC AGG A | Initial denaturation of 2 min at 94 °C, followed by 30 cycles at 94 °C for 30 s at, 52 °C for 1 min and 72 °C for 30 s, and final elongation at 72 °C for 10 min. | (*Wei et al., 2018*) |
| *nheABC* | F-AAG CIG CTC TTC GIA TTC R-ITI GTT GAA ATA AGC TGT GG | Initial denaturation of 5 min at 94 °C, followed by 30 cycles at 94 °C for 30 s, 49 °C for 1 min and at 72 °C for 1 min, and final elongation at 72 °C for 5 min. | (*Ehling-Schulz et al., 2006*) |
| *hblABD* | F-GTA AAT TAI GAT GAI CAA TTT C R-AGA ATA GGC ATT CAT AGA TT | | |
| *ces* | F-TTG TTG GAA TTG TCG CAG AG R-GTA AGC GGA CCT GTC TGT AAC AAC | Initial denaturation of 2 min at 94 °C, followed by 30 cycles at 94 °C for 30 s at, 52 °C for 1 min and 72 °C for 30 s and a final elongation at 72 °C for 10 min. | |
| *cytK-plcR* | P1-CAA AAC TCT ATG CAA TTA TGC AT P3-ACC AGT TGT ATT AAT AAC GGC AAT C | Initial denaturation of 2 min at 94 °C, followed by 30 cycles at 94 °C for 30 s at, 52 °C for 1 min and 72 °C for 30 s, and final elongation at 72 °C for 10 min. | (*Oltuszak-Walczak & Walczak, 2013*) |
| *tasA* | F-AGC AGC TTT AGT TGG TGG AG R-GTA ACT TAT CGC CTT GGA ATTG | Initial denaturation of 5 min at 94 °C, followed by 40 cycles at 94 °C for 30 s, 59 °C for 45 s and 72 °C for 45 s, and final elongation at 72 °C for 5 min. | (*Caro-Astorga et al., 2015*) |
| *sipW* | F-AGA TAA TTA GCA ACG CGA TCTC R-AGA AAT AGC GGA ATA ACC AAGC | Initial denaturation of 5 min at 94 °C, followed by 40 cycles at 94 °C for 30 s, 54 °C for 45 s, and a final elongation at 72 °C for 5 min. | |

**Note:**
I: inosine.

and then placed on ice. After centrifugation, the supernatant was used as template for the molecular identification and toxin profile (*Adame-Gómez et al., 2020*).

The differentiation of *B. cereus* group was targeted on *gyrB* gene, and the toxin gene profiles from *nheABC*, *hblABD*, *ces*, and *cytK* gene. The reaction mixes (25 μL) contained the following: 12.5 μL of REDTaq Ready Mix DNA polymerase (Sigma-Aldrich, St. Louis, MI, USA), 11 μL of sterile Milli-Q water, 0.5 μL of the genomic DNA template (concentration about 10–20 ng/μL), and 0.02 μM of each primer. The oligonucleotides and the conditions of the different PCRs are shown in Table 1. The PCR products were analyzed by 2% agarose gel electrophoresis at 80 V for 120 min the gels were stained with Midori Green (Nippon Genetics, Düren, Germany) and visualized on an LED transilluminator. The *B. cereus* ATCC14579 strain was used as a positive control for amplifying the *gyrB*, *hbl*, *plcR-cytK*, *sipW* and *tasA* genes. Strain *B. cereus* BC133 was used as a positive control for *nhe* amplification. This strain was previously isolated and characterized in the laboratory from infant formula powdered. The strain *B. subtilis* BC001 was used as a negative control for *nhe*, *hbl*, *plcR* and *cytk*. The BC001 strain was isolated from infant formula powdered and previously characterized in the laboratory.

## Biofilm production

Biofilms were generated in glass tubes with 3 mL of brain heart infusion (BHI) broth (Bioxon, Mexico City, Mexico). The broths were inoculated with 0.3 mL of a previous bacterial culture of 24 h. The tubes were incubated at 30 °C for 48 h under static conditions, once the incubation time had elapsed, the microbial growth was evaluated at an absorbance of 600 nm. After the medium was removed, the tubes were washed three times with phosphate buffered saline (PBS) (Life Technologies, Carlsbad, CA, USA). Adhered biofilms were stained with a 0.1% crystal violet solution for (BD Difco, Franklin Lakes, NJ, USA) 30 min. The glass tubes were washed three times with PBS and incubated with 70% alcohol for 30 min to release the crystal violet attached to the biofilms. The solubilized crystal violet was quantified by absorbance at 595 nm (Castelijn et al., 2012). We used the B. cereus strain ATCC14579 as a positive control and the culture medium without inoculum was used as a negative control.

## Determination of genes related to biofilm production in B. cereus

The sipW and tasA genes were amplified by PCR. The final reaction mix contains 0.2 μM of each dNTP, 3 μM MgCl2, 0.2 mM of the oligonucleotides, 1 U of Taq DNA polymerase (Ampliqon, Odense, Denmark), 5 μL of 10× buffer, and 100 ng of DNA as template. The PCR conditions and oligonucleotides used were described by Caro-Astorga et al. (2015) and are shown in Table 1. The PCR products were analyzed by 2% agarose gel electrophoresis at 80 V for 60 min, the Gels were stained with Midori Green (Nippon Genetics, Düren, Germany) and visualized on an LED transilluminator. The B. cereus ATCC14579 strain was used as a positive control.

## Enzymatic profile

The capacity of the strains to degrade casein was tested on soy agar plates supplemented with 15% skimmed milk and incubated at 30 °C for 24 h, the presence of clear zones around the colonies indicates hydrolysis of the casein (Claus & Berkeley, 1986).

The ability to hydrolyze starch was determined on soy agar plates with 0.25% starch. The inoculated plates were incubated at 30 °C for 3 days. A drop of Lugol's iodine solution was placed on the edge of the colonies obtained after incubation. Amylolytic properties were confirmed by the change from the blue to the yellow color of the reagent (Claus & Berkeley, 1986).

## Analysis of presumed psychotropic properties

All strains were screened for their psychotropic growth at 7 °C ± 1 °C. They were plated on Tryptone Soya Agar (Oxoid, Basingstoke, United Kingdom) and incubated at 7 °C ± 1 °C for 7 days. Growth was checked after 24, 48, 72 h, and 1 week (Berthold-Pluta et al., 2019).

## Determination of sporulation rates

Before removing the BHI broth from the biofilm production assay, 100 μL of culture was taken and resuspended in 900 μL of sterile distilled water, dilutions were made by factor 10, of which 100 μL were dispersed on BHI agar for calculate number of cells. The initial

dilution was subjected to 80 °C for 10 min to eliminate vegetative cells; once the time had elapsed, dilutions were made by factor 10, of which 100 μL were dispersed on BHI agar for calculate spores. The both plates were incubated at 30 °C for 48 h, with the respective colony-forming units (CFU) counting. The experiments were repeated in triplicate (*Hayrapetyan et al., 2015*).

### Genetic diversity of *B. cereus* strains

The 17 strains of *B. cereus* were selected to determine the diversity among the iso-lates from PCR of repetitive elements using oligonucleotide (GTG) 5 (*De Jonghe et al., 2008*). The electro-phoretic patterns obtained were analyzed using the NTSYs software. Genetic similarity was determined *via* the DICE coefficient and represented in a dendrogram from hierarchical clustering. A coefficient of DICE >0.8 was used to group the strains into a cluster.

### Statistical analysis

The results represent the average of at least two independent experiments. All continuous variables were expressed as mean ± standard deviation. The sporulation rate and biofilm formation by *B. cereus* strains were compared using one-way analysis of variance (ANOVA) with a Bonferroni correction (Bonferroni *post hoc* test). For all tests, a *p*-value of 0.05 was assumed to be the minimum level of significance.

## RESULTS

The prevalence of *B. cereus* in vegetables by molecular identification of the *gyrB* gene was 20% (13/65); A greater number of strains were recovered (17 strains) because more than one strain with different colonial morphology characteristics was isolated in three samples (samples 57, 58 and 62). No suspicious colony morphologies were found for *B. mycoides* (opaque colonies that are characteristically hairy in appearance, often referred to as rhizoid). The cry gene was used to discriminate *B. thuringiensis*; however, all strains were negative. *B. cereus* strains were isolated only in coriander from municipal markets and not in coriander from supermarkets. In this study, all samples exceeded $10^5$ CFU/g.

The most common enterotoxin gene was *nhe* [76.4%, (13/17)], followed by *cytK* [82.3%, (14/17)]; in this study, no strains with genes for the HBL toxin and cereulide toxin were found (Fig. 1). In the case of genes related to biofilms, the frequency was low for *sipW* [5.8%, (1/17)] and *tasA* [11.7%, (2/17)]. Sporulation rates were 80% for most strains, while biofilm production was low in all strains (Fig. 2). Only two strains negative for genes for toxins and biofilm-related genes were found in this study. The most common enterotoxigenic profile was the combination of the *nhe* and *cytK* genes, this profile was found in 12 strains; two strains were found that were only positive for the *cytK* gene, one strain only had the *nhe* gene. Of the genes related to biofilms, only one strain presented the two genes of interest (*sipW–tasA*), and one strain showed only the *tasA* gene. Also, a high frequency of protease production [47%, (8/17)] and low amylase production [17.6%, (3/17)] was observed in the strains. Most of the strains were able to grow in the first three

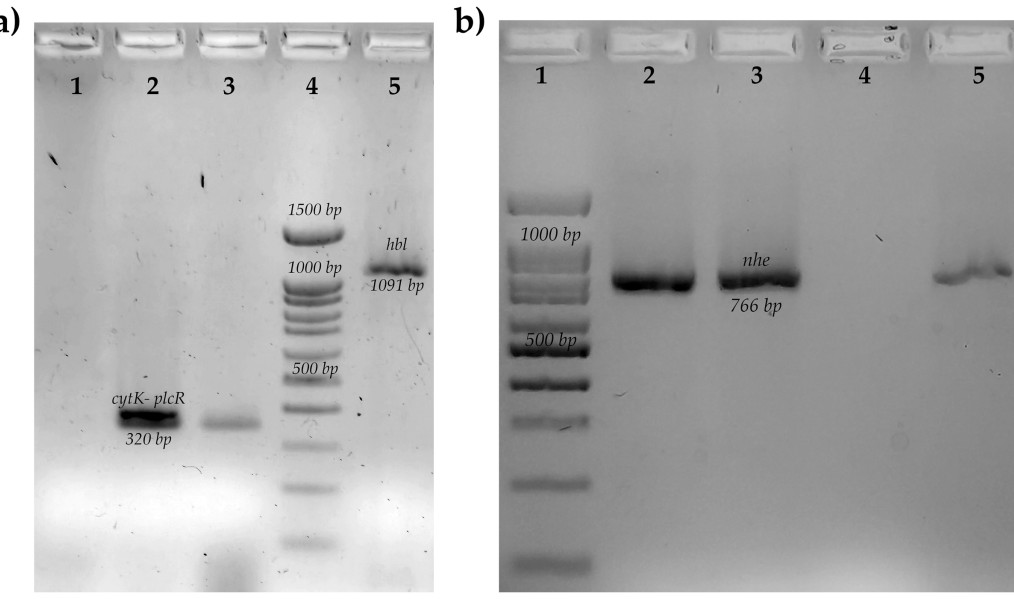

**Figure 1 Molecular identification of toxins genes of _B. cereus_ from coriander isolation.**
(A) Amplification of cytotoxin K gene (cytk-PlcR, 320 bp) lane 1: Control negative _B. subtilis_, lane 2: positive control _B. cereus_ ATCC14579, lane 3: BC006, lane 4: molecular weight market of 100 bp, lane 5: amplification of hemolytic BL gene (hbl, 1,091 bp) positive control _B. cereus_ ATCC14579. (B) Amplification of no-hemolytic enterotoxin gene (nhe, 766 bp) lane 1: molecular weight market of 100 bp, lane 2: positive control _B. cereus_ BC133, lanes 3,5: BC006 and BC030, lane 4: negative control _B. subtilis_.

days at temperatures of 7 °C [88.2% (15/17)]. Of the strains that grew after one day of incubation, 80% (4/5) were positive for an enterotoxin gene (Table 2).

As for genetic diversity, we observed that strains isolated from the same market, but different vegetable retailers are grouped into a cluster (BC010 and BC034, BC032 and BC040; y BC058C, B064B); however, we also observed isolated strains from different markets that are grouped in clusters (BC039 and BC064; BC030 and BC050). Regarding enzymatic and virulence profiles, the strains are distributed throughout the dendrogram; in the case of enterotoxin-negative strains, they are grouped in a cluster (BC010 and BC034). The strains isolated from the same coriander sample do not present the same genetic profile and are not grouped in the same cluster (BC057, BC057B/ BC058, BC058B, BC058C). Strains isolated from samples from the same vendor but purchased in different weeks did not cluster (BC006/ BC010) (BC039/ BC041) (Fig. 3).

## DISCUSSION

_B. cereus_ is a microorganism that is usually found in the soil due to the production of spores (_Vilain et al., 2006_) and the symbiotic relationships with other organisms such as arthropods (_Mukhopadhyay & Chatterjee, 2016_); therefore, the contamination of vegetables by this microorganism is evident, from pre-harvest, harvest, and post-harvest (_Maffei et al., 2016_). For the above, the isolation and characterization of _B. cereus_ in coriander are important because it is a high vegetable consumption in Mexico. A country that is also actively involved in exporting this product.

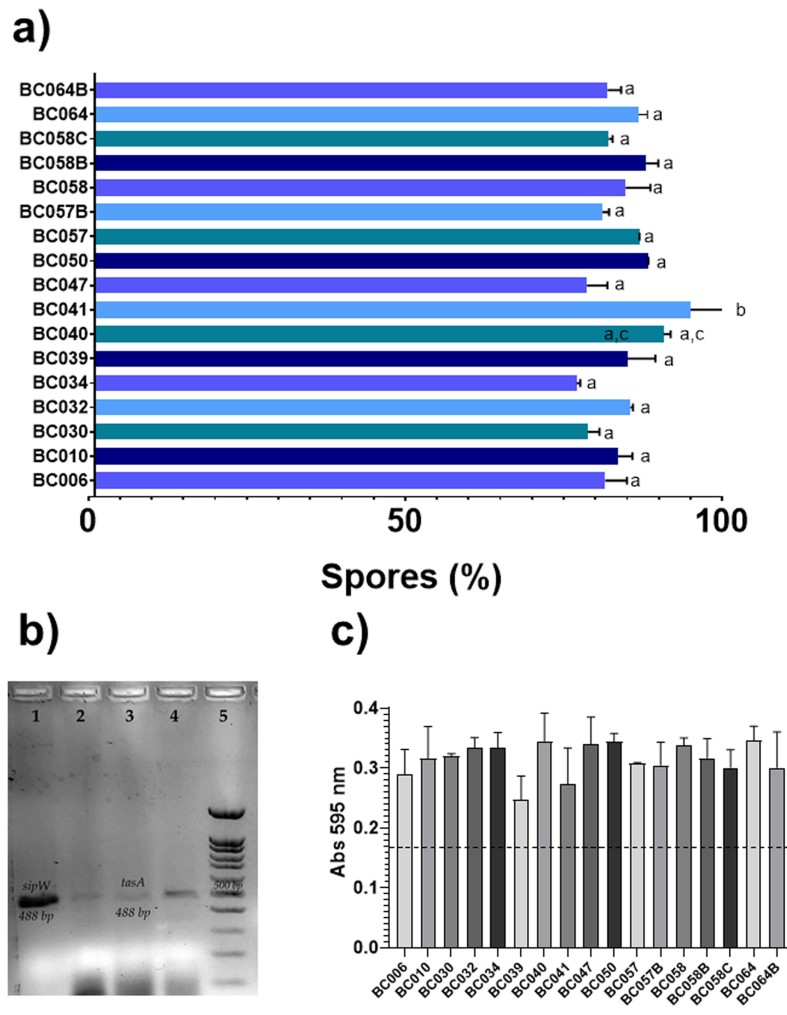

**Figure 2 Determination of sporulation rates, biofilm production and genes related to biofilms of B cereus strains from coriander.** Determination of sporulation rates, biofilm production and genes related to biofilms of B cereus strains from coriander. (A) Sporulation rates of all *B. cereus* strain. The bars are the means of % spores. The same letter indicates that there are no statistically significant differences. (a and c, $p = 0.05$) (a and b, $p = 0.0001$). In the X axis are the % spores and, in the Y axis is the strains. ANOVA *post hoc* Bonferroni. The experiments were performed in duplicate. (B) Agarose gel electrophoresis of the PCR products of tasA and sipW genes. Lane 1: positive control for sipW (488 bp) *B. cereus* ATCC14579, lane 2: sipW+ strain, lane 3: positive control for tasA (488 bp) *B. cereus* ATCC14579, lane 4: tasA+ strain, lane 5: molecular weight market of 100 bp. (C) Determination of biofilm production all *B. cereus* strains. The bars are the means of absorbance of crystal violet. The strains are indicated in the X axis, and abdorbance to 595nm is indicated by the Y axis. ANOVA. The experiments were performed in duplicate.

The frequency of *B. cereus* in coriander is like that reported in the analysis of vegetables in France (*Glasset et al., 2016*) and Korea (*Kim et al., 2016*); however, it differs from other countries, even where coriander samples were processed, including Mexico, where the prevalence is three times higher (*Flores-Urbán et al., 2014*; *Yu et al., 2020*). One possible explanation is that we only included the leaves in our study and not the stems. It is important to further analyze the frequency of this organism even with the methodological difference. It has shown that the microbiological composition of the soil (the main source

Table 2 Genetic, enzymatic and psychrophilic profiles of *B. cereus* strains.

| Strain | Toxin profile | | | | Biofilm related genes | | Enzymes | | Psychrophilia (days) |
|---|---|---|---|---|---|---|---|---|---|
| | *nhe* | *hbl* | *cytK* | *ces* | *sipW* | *tasA* | Amylases | Proteases | |
| BC006 | + | − | + | − | − | − | + | − | 3 |
| BC010 | − | − | − | − | − | − | − | − | 1 |
| BC030 | + | − | + | − | + | + | + | − | 2 |
| BC032 | − | − | + | − | − | − | − | − | 1 |
| BC034 | − | − | − | − | − | − | − | − | 3 |
| BC039 | + | − | − | − | − | − | − | + | 2 |
| BC040 | + | − | + | − | − | − | − | − | 3 |
| BC041 | − | − | + | − | − | − | − | − | 3 |
| BC047 | + | − | + | − | − | − | − | − | 3 |
| BC050 | + | − | + | − | − | − | − | + | 3 |
| BC057 | + | − | + | − | − | + | − | − | 1 |
| BC057B | + | − | + | − | − | − | − | + | 1 |
| BC058 | + | − | + | − | − | − | + | + | 3 |
| BC058B | + | − | + | − | − | − | − | + | 2 |
| BC058C | + | − | + | − | − | − | − | + | − |
| BC064 | + | − | + | − | − | − | − | + | − |
| BC064B | + | − | + | − | − | − | − | + | 1 |

Note:
BC, *Bacillus cereus*.

of contamination of vegetables by *B. cereus*) can vary according to various chemical agents or biological that can be added as part of the harvest.

For example, it has been observed that green manures favor the presence of various bacterial genera (*Longa et al., 2017*) or the presence of pesticides; in this sense, even strains of *Bacillus* with degradation capabilities of the pesticide have been isolated (*Jiang et al., 2019*) or with high resistance to heavy metals (*Ayangbenro & Babalola, 2020*). Also, by collecting soil samples from four farms enriched with *Clostridium* strains, the inhibition of *B. cereus* and *B. mycoides* has been demonstrated. Metabolites extracted from the soil found that the metabolite that is attributed to antimicrobial activity against strains of *B. cereus* was isocaproic-hydroxy-2- (*Pahalagedara et al., 2020*).

And not only would soil explain the differential distribution of *B. cereus* strains in vegetables, but also the characteristics of the vegetables themselves. For example, green leafy vegetables are rich in cellulose in stems and leaves; However, bulb-shaped vegetables are rich in starch (*Hurtado-Barroso et al., 2019*). This carbohydrate can be metabolized by strains of *B. cereus* that possess the enzyme amylase. The above could explain the differences in variations in the frequency of *B. cereus* in different vegetables.

*B. cereus* is a microorganism commonly found in soil; therefore, it is not surprising that it is widely distributed in vegetable-growing areas. Consequently, it is important to detect this microorganism from the pre-harvest and not only during the sale. The latter is to

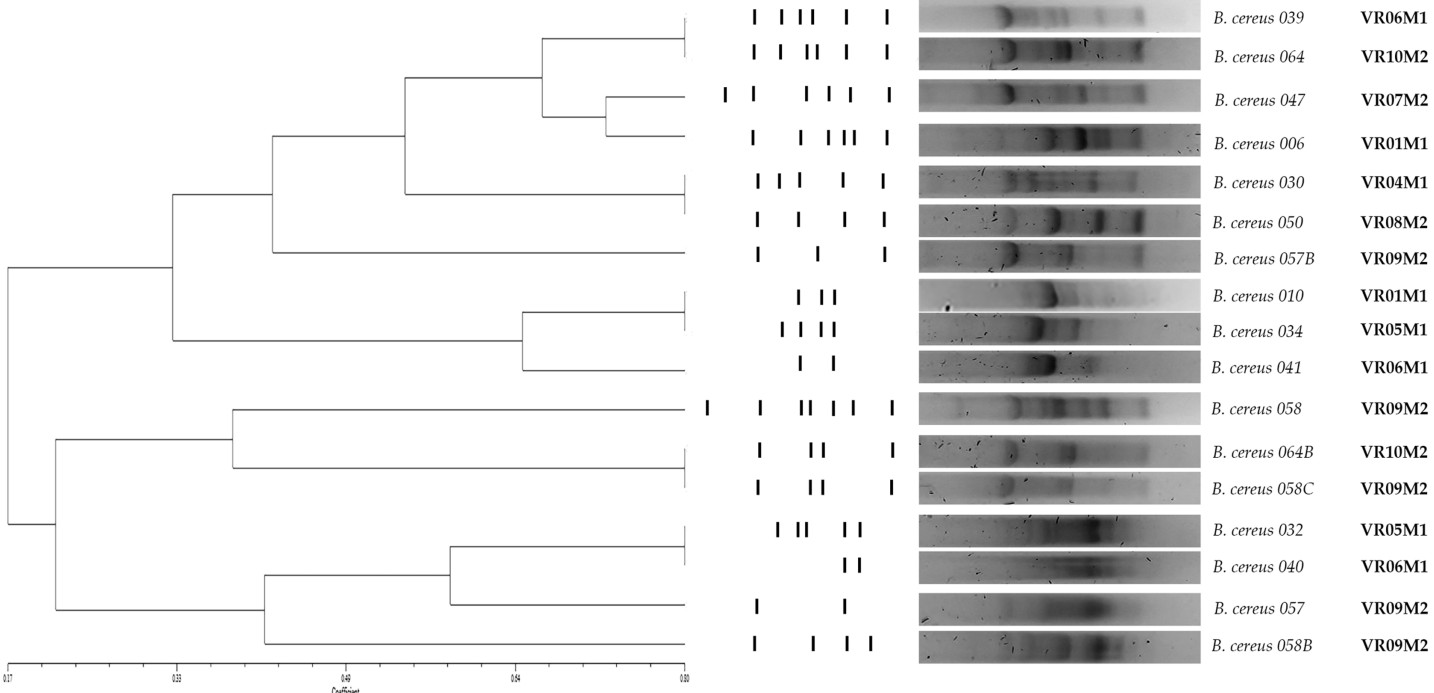

**Figure 3 Dendrogram obtained by GTG5 of strains of the *B. cereus* group.** Seventeen strains from coriander were positive to gyrB which were characterized and grouped into 12 main groups and four clones. DICE test was used, and the strains were grouped by hierarchy clustering. A coefficient of DICE >0.8 was used to group the strains into a cluster. VR is the vegetable retailer number and M is the market number.

identify other possible routes of contamination and strategies for the control of the microorganism.

In this study, all samples exceeded $10^5$ CFU/g. In this sense, the FDA estimates that infectious doses above $10^6$ CFU/g are needed to cause disease (*Tallent et al., 2021*). For one food poisoning to occur, it depends not only on the infectious dose but also on the characteristics of the microorganism (*Granum & Lund, 2006*; *EFSA Panel on Biological Hazards (BIOHAZ), 2016*). Therefore, lower numbers of CFU/g strains, such as those found in this study, should not be estimated. Other regulatory institutions establish limits for this microorganism in ready-to-eat foods (vegetables could fall into this group). Food microbiological guidelines in Hong Kong, China state that the amounts of *B. cereus* in ready-to-eat foods must be between $10^3$–$10^5$ CFU/g to be considered acceptable (*Food & Environmental Hygiene Department, 2014*).

Meanwhile, the microbiological standards of the United Kingdom establish that in ready-to-eat foods, the limit to consider them acceptable is $10^3$–$10^5$ CFU/g (*Health Protection Agency, 2009*). Finally, the Australian and New Zealand legislation for ready-to-eat foods considers a *B. cereus* level of $10^2$–$10^3$ CFU/g acceptable (*New South Wales Food Authority, 2009*). Unfortunately, in Mexico, the current sanitary legislation does not consider *B. cereus* in ready-to-eat foods such as vegetables or other products, highlighting the importance of this type of study to demonstrate the circulation of this microorganism. Furthermore, it is essential to consider that most of the strains in this

study are psychrophilic, which allows them to continue growing at refrigeration temperatures. The above would increase the probability of consuming products with a high number of CFUs close to the infectious dose. On the other hand, cleaning these products has been considered a strategy to reduce the microbial load (*Hilgren & Salverda, 2000*). In this sense, we must emphasize that the production of biofilms by this microorganism has been described, which favors not only its environmental resistance but also its resistance to disinfectant agents (*Hussain, Kwon & Oh, 2018*; *Majed et al., 2016*).

Regarding the enterotoxigenic profile, in this study, a high frequency of genes coding for NHE and CYTK enterotoxin was found, which is like that reported in Korea (*Park et al., 2018*) and France (*Glasset et al., 2016*) in vegetables and in general with various studies in other food matrices (*Ankolekar, Rahmati & Labbé, 2009*; *Chaves, Cavados & Vivoni, 2012*; *Chon et al., 2012*; *Guinebretière, Broussolle & Nguyen-The, 2002*; *Hansen & Hendriksen, 2001*; *Ouoba, Thorsen & Varnam, 2008*). It is important to note that a significant variety of studies in vegetables report strains with HBL enterotoxin genes (*Flores-Urbán et al., 2014*; *Longa et al., 2017*; *Yu et al., 2020*; *Drewnowska et al., 2020*; *Gdoura-Ben Amor et al., 2019*; *Zervas et al., 2020*); however, in this study, no strains with genes for the HBL toxin were found.

In addition, the presence of the *ces* gene that codes for cereulide was not found, which has been previously reported (*Altayar & Sutherland, 2006*; *Chon et al., 2012*; *Ouoba, Thorsen & Varnam, 2008*). We can also attribute it to all strains included in the study being psychrophilic and reported that the emetic strains could not grow at lower temperatures than non-emetic strains (*Carlin et al., 2006*). Regarding distribution, high frequencies of positive strains for *ces* genes have been reported more frequently in some European countries (*Glasset et al., 2016*; *Berthold-Pluta et al., 2019*; *Gdoura-Ben Amor et al., 2019*), which coincides with a higher rate of emetic syndrome cases in this region (*Eurosurveillance Editorial Team, 2013*). In this last point, it has also been observed that the distribution of psychrophilic and mesophilic strains and virulence factors is determined by the geographical area, considering that the environment plays an important role in the distribution of the microorganism (*Drewnowska et al., 2020*).

It described that *B. cereus* might produce biofilms on roots of plants, and consumption of the roots from various nematodes could promote dispersion of the high part of the plant (*Majed et al., 2016*); besides roots, it described that could produce biofilms on the surface of vegetables (*Ehling-Schulz, Fricker & Scherer, 2004*; *Tatsika et al., 2019*); which could explain the presence of this microorganism in vegetables. However, in this study, the production of biofilms was low, and as the frequency of the *sipW* and *tasA* genes. In a previous study, we found a differential distribution of the *sipW* and *tasA* genes according to the origin of the *B. cereus* strains, finding a high frequency in strains from eggs compared to strains from other foods (*Adame-Gómez et al., 2020*).

Therefore, it is considered that another mechanism that could explain the presence of *B. cereus* in coriander is the production of spores. We determine the sporulation rates and, we find that all strains produce *in vitro* rates of sporulation around 80%. We consider that this could be the reason for its presence or persistence in vegetables. We are remarking that spores are resistant to high and low temperatures, desiccation, dis-infecting agents,

ionizing radiation, and ultraviolet light. The spores allow its wide distribution in the soil, water, air, plants, and animals; and therefore, in food products (*Vidic et al., 2020*).

The sporulation of *B. cereus* as the main mechanism of persistence in vegetables has been shown previously, in which the *B. cereus* strain DSM2302 cannot survive on melon leaves. The *B. cereus* DSM 2302 strain cannot sporulate due to the lack of genes for the intermediate and late sporulation phases; however, it does have genes related to biofilm production. Sporulation protects the strains against adverse environmental conditions in the plant. For example, it has been described that carbon and nitrogen sources are scarce in melon leaves, and there is a high level of secondary metabolites such as flavonoids (*Antequera-Gómez et al., 2021*).

The evaluation of genetic diversity by the $(GTG)_5$ genotyping showed that 12 strains grouped into at least six clusters. Four clusters contain strains from different vendors but the same market. The rest include strains from different markets. On the one hand, this study evidences the wide genetic diversity of *B. cereus* strains previously described in other food matrices with this same technique (*Sánchez-Chica et al., 2021a*; *Sánchez-Chica et al., 2021b*; *Samapundo et al., 2011*; *Samapundo et al., 2014*). On the other hand, the concentration of coriander producers in the markets coming from different locations near the region, with possible differences in pre-harvest, harvest, and post-harvest stages, could explain this genetic diversity.

Regarding the safety of coriander, outbreaks of cyclosporiasis, as well as *Salmonella* have been reported in the USA associated with the consumption of this product of Mexican origin (*Campbell et al., 2001*; *Abanyie et al., 2015*). Therefore, it is important to continue the search for pathogens such as *C. cayetanensis* and *Salmonella* in coriander but also highlight the presence of other food-associated pathogens such as *B. cereus*.

In the coriander marketed in southwestern Mexico, different strains of *B. cereus* were found with genes associated with the production of diarrheal toxins, which is why it is important to search for strategies to control the microorganism and its legislation, as well as other microorganisms such as *Salmonella*.

## CONCLUSIONS

- The prevalence of *B. cereus* in vegetables by molecular identification of the *gyrB* gene was 20%
- The most common enterotoxin gene was *nhe* [76.4%, (13/17)], followed by *cytK* [82.3%, (14/17)]
- In the case of genes related to biofilms, the frequency was low for *sipW* [5.8%, (1/17)] and *tasA* [11.7%, (2/17)]
- The sporulation rates could be responsible for the presence of *B. cereus* in vegetables due to the resist characteristic of the spores previously reported.
- Further analyses are necessary for identification and molecular characterization of other strains of *B. cereus* in vegetables for human consumption.

### Funding

Daniel Alexander Castulo-Arcos received a fellowship from the National Science and Technology Council as support for this work. The funders had no role in study design, data collection and analysis, decision to publish, or preparation of the manuscript.

### Grant Disclosures

The following grant information was disclosed by the authors:
National Science and Technology Council.

### Competing Interests

The authors declare that they have no competing interests.

### Author Contributions

- Daniel Alexander Castulo-Arcos performed the experiments, prepared figures and/or tables, and approved the final draft.
- Roberto Adame-Gómez performed the experiments, prepared figures and/or tables, and approved the final draft.
- Natividad Castro-Alarcón conceived and designed the experiments, prepared figures and/or tables, and approved the final draft.
- Aketzalli Galán-Luciano performed the experiments, authored or reviewed drafts of the article, discussion, and approved the final draft.
- María Cristina Santiago Dionisio conceived and designed the experiments, authored or reviewed drafts of the article, and approved the final draft.
- Marco A. Leyva-Vázquez analyzed the data, prepared figures and/or tables, authored or reviewed drafts of the article, and approved the final draft.
- Jose-Humberto Perez-Olais analyzed the data, authored or reviewed drafts of the article, and approved the final draft.
- Jeiry Toribio-Jiménez analyzed the data, authored or reviewed drafts of the article, and approved the final draft.
- Arturo Ramirez-Peralta conceived and designed the experiments, analyzed the data, prepared figures and/or tables, authored or reviewed drafts of the article, and approved the final draft.

### Data Availability

The raw measurements of sporulation rates and biofilm are available in the Supplemental Files.

### Supplemental Information

Supplemental information for this article can be found online at http://dx.doi.org/10.7717/peerj.13667#supplemental-information.

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
