# Peer review of "Genetic diversity of enterotoxigenic Bacillus cereus strains in coriander in southwestern Mexico"

_PeerJ, doi:10.7717/peerj.13667_

## Round 0.1 · original submission · Major Revisions

Please address all concerns raised by both reviewers, in particular, the comments from Reviewer 1, who suggest additional experiments.

Reviewer 1 ·

Basic reporting

This manuscript, entitled “Genetic diversity of enterotoxigenic Bacillus cereus strains in coriander in southwestern Mexico”, had described the toxigenic profile, biofilm production, genes associated with the production of biofilms, sporulation rates, enzymatic profile, psychotropic properties, and genetic diversity of B. cereus strains from coriander in southwestern Mexico. Due to lack of significance and novelty, my suggestion is rejection.

Experimental design

First and most importantly, the samples were collected from a relatively small area within a short period (six-week study period). Also, only 60 coriander samples and 17 B. cereus strains were recovered. The results yield from such samples have limited guidance and significance.
Secondly, only toxigenic profile, biofilm production, genes associated with the production of biofilms, sporulation rates, enzymatic profile, psychotropic properties, and genetic diversity were included in the study. Such phenotype screening is pre-experimental studies upon further in-depth molecular mechanism investigation. The data are insufficient to be published as a scientific manuscript unless there’s novel strain types and genes identified. However, in the current study, none of such types were identified.

Validity of the findings

Thirdly, the results and discussion parts were short and dull, the data presentation is poor. Due to the lack of significance data to be discussed, I suggest the authors to perform more in-depth study.

Reviewer 2 ·

Basic reporting

The manuscript written by Cástulo-Arcos et al. reports some interesting results. In this work, the authors have discovered that Bacillus cereus strains produce a low amount of biofilm and produce rates sporulation around 80%.

Experimental design

In general, this paper is clearly laid out, well planed and easy to read. The experiments are well designed and appropriate controls are presented.

Validity of the findings

The novelty and significance of the manuscript were not highlighted, please modify the paper more clearly.

Additional comments

Manuscript Title: Genetic diversity of enterotoxigenic Bacillus cereus strains in coriander in southwestern Mexico
Manuscript ID: #71248
The manuscript written by Cástulo-Arcos et al. reports some interesting results. In this work, the authors have discovered that Bacillus cereus strains produce a low amount of biofilm and produce rates sporulation around 80%.
Some specific suggestions or questions are listed below:
1.Abstract: Abstract should be written more precisely and explain novelty of this work.
2.Introduction: Introduction is easy to read but needs a little completed. Bacillus strains are well known for their metabolic capability and environmental versatility as well as for their ability to manage bacterial and fungal pathogens infecting crop plants. Authors should add more information into this section and cite the recent research into the field based on the recent literature (such as doi: 10.3390/plants11030457; doi: doi: 10.3390/microorganisms10020365; 10.3390/microorganisms9122511; doi: 10.1016/j.jwpe.2020.101712; doi: 10.3390/plants10112342; doi:10.3390/microorganisms8020223; doi: 10.1016/j.postharvbio.2013.10.004; doi: 10.1007/s00253-014-6164-y; doi: 10.1016/j.biortech.2012.01.106; doi: 10.1016/j.chemosphere.2022.133609; doi: 10.1021/jf404908j; doi: 10.3389/fbioe.2020.570307).
3.Introduction: Coriandrum sativum, Please check throughout the manuscript that abbreviations/acronyms are defined the first time they appear in each of three sections: the abstract; the main text; the first figure or table.
4.Introduction: In addition, this part should focus on the research progress related to the topic and emphasize the innovation of this research. However, the novelty and significance of the topic were not highlighted, please modify the introduction more clearly.
5.Material and Methods: please add relative references to support all the methods.
6.Results: This section is too simple. Please add all the results in details.
7.Results: Figure 3 is not clear. Please re-draw the figure and provide high quality one.
8.Discussion: Authors should add more depth discussion into this section and cite the recent research into the field.
9.Conclusions: Authors can add and revise the Conclusions section for the better understanding of the topic and its future research.
References: Many of the references have been superceded and more modern ones are required

---

## Round 0.2 · accepted · Accept

The manuscript was significantly modified, following the Reviewers' comments. As a consequence, the manuscript content is suitable for publication in Peer Journal.

---

## Author Rebuttal · Round 0.2

**Universidad Autónoma de Guerrero**
**Facultad de Ciencias Químico-Biológicas**
**Laboratorio de Investigación en Patometabolismo Microbiano**
Av. Lázaro Cárdenas s/n. Ciudad Universitaria, Chilpancingo, Gro, México. C.P. 39070
Tel. Fax (01747)47 25503. Tel 4719310 ext. 3636

May 22th, 2021

Dear Editors

We thank the reviewers for their generous comments on the manuscript and have edited the manuscript to address their concerns.

The review comments are in black, while the author response is in blue

We believe that the manuscript is now suitable for publication in PeerJ

Ph.D. Arturo Ramírez Peralta

Professor, Department of Chemistry

Universidad Autonoma de Guerrero

Chilpancingo, Guerrero, Mexico

[Figure]

The review comments are in black, while the author response is in blue

Reviewer 1 (Anonymous)

Basic reporting

This manuscript, entitled "Genetic diversity of enterotoxigenic Bacillus cereus strains in coriander in southwestern Mexico", had described the toxigenic profile, biofilm production, genes associated with the production of biofilms, sporulation rates, enzymatic profile, psychotropic properties, and genetic diversity of B. cereus strains from coriander in southwestern Mexico. Due to lack of significance and novelty, my suggestion is rejection.

*In Mexico, there are few studies about the circulation of Bacillus cereus strains. Some of these studies have been carried out by our laboratory.*

*Adame-Gómez, R., Muñoz-Barrios, S., Castro-Alarcón, N., Leyva-Vázquez, M.-A., Toribio-Jiménez, J., & Ramírez-Peralta, A. (2020). Prevalence of the Strains of Bacillus cereus Group in Artisanal Mexican Cheese. Foodborne Pathogens and Disease, 17(1), 8-14. https://doi.org/10.1089/fpd.2019.2673*

*Flores-Urbán, K. A., Natividad-Bonifacio, I., Vázquez-Quiñones, C. R., Vázquez-Salinas, C., & Quiñones-Ramírez, E. I. (2014). Detection of Toxigenic Bacillus cereus Strains Isolated from Vegetables in Mexico City. Journal of Food Protection, 77(12), 2144-2147. https://doi.org/10.4315/0362-028X.JFP-13-479*

*Hernández, A. G. C., Ortiz, V. G., Gómez, J. L. A., López, M. Á. R., Morales, J. A. R., Macías, A. F., Hidalgo, E. Á., Ramírez, J. N., Gallardo, F. J. F., Gutiérrez, M. C. G., Gómez, S. R., Jones, G. H., Flores, J. L. H., & Guillén, J. C. (2021). Detection of Bacillus cereus sensu lato Isolates Posing Potential Health Risks in Mexican Chili Powder. Microorganisms, 9(11), 2226. https://doi.org/10.3390/microorganisms9112226*

*Cruz-Facundo, I., Adame-Gómez, R., Vences-Velázquez, A., Rodríguez-Bataz, E., Muñoz-Barrios, S., Pérez-Oláis, J., & Ramírez-Peralta, A. (2022). Bacillus Cereus in Eggshell: Enterotoxigenic Profiles and Biofilm Production. Brazilian Journal of Poultry Science, 24(2), eRBCA-2021-1535. https://doi.org/10.1590/1806-9061-2021-1535*

*Adame-Gomez, R., Castro Alarcón, N., Vences-Velázquez, A., Rodríguez-Bataz, E., Santiago-Dionisio, M. C., & Ramírez-Peralta, A. (2019). Prevalencia de cepas del grupo de Bacillus cereus productoras de biopelicula en helados comercializados en México. https://doi.org/10.5281/ZENODO.3520530*

*As for vegetables, also the number of studies is limited.*

*Flores-Urbán, K. A., Natividad-Bonifacio, I., Vázquez-Quiñones, C. R., Vázquez-Salinas, C., & Quiñones-Ramírez, E. I. (2014). Detection of Toxigenic Bacillus cereus Strains Isolated from Vegetables in Mexico City. Journal of Food Protection, 77(12), 2144-2147. https://doi.org/10.4315/0362-028X.JFP-13-479*

*Hernández, A. G. C., Ortiz, V. G., Gómez, J. L. A., López, M. Á. R., Morales, J. A. R., Macías, A. F., Hidalgo, E. Á., Ramírez, J. N., Gallardo, F. J. F., Gutiérrez, M. C. G., Gómez, S. R., Jones, G. H., Flores, J. L. H., & Guillén, J. C. (2021). Detection of Bacillus cereus sensu lato Isolates Posing Potential Health Risks in Mexican Chili Powder. Microorganisms, 9(11), 2226. https://doi.org/10.3390/microorganisms9112226*

*Therefore, we consider that it would be the first point for which our study is important. This information is reflected in the introduction (lines 99-101). Mexico has positioned itself as an exporter of this product, so it is also important to have updated information about this product. We mention the above in the discussion (lines 257-259). We believe that updating the information on this microorganism will allow this microorganism to be included in the country's health legislation in the future, considering, for example, the CFUs found in these samples. This information is in the results section and the discussion (lines 220 -221, 299-302)*

Experimental design

First and most importantly, the samples were collected from a relatively small area within a short period (six-week study period). Also, only 60 coriander samples and 17 B. cereus strains were recovered. The results yield from such samples have limited guidance and significance.

*Many studies on the circulation of strains of B. cereus are cross-sectional. This six-week follow-up study allowed us to identify the genetic diversity of the B. cereus strains. For example, strain B006 is not the same as strain B010, and these strains are isolated from cilantro from the same vendor but on a different week (line's 246- 248, figure 3). The above could indicate that every time the seller arrives at the market, he introduces new strains of B. cereus from cilantro. On the other hand, even strains isolated from the same coriander sample are not the same. The above allows us to conclude the broad genetic diversity of B. cereus in this product, like other food matrices. This information is found in the discussion part (lines 361-369). Despite being only six weeks long, this follow-up study allowed us to draw this important conclusion.*

Secondly, only toxigenic profile, biofilm production, genes associated with the production of biofilms, sporulation rates, enzymatic profile, psychotropic properties, and genetic diversity were included in the study. Such phenotype screening is pre-experimental studies upon further in-depth molecular mechanism investigation. The data are insufficient to be published as a scientific manuscript unless there's novel strain types and genes identified. However, in the current study, none of such types were identified.

*We consider that the genus Bacillus is indeed versatile. Currently, many strains have stood out for their bioremediation or bioinsecticide capacity, which is reflected in the introduction (lines 65- 79). Therefore, it is important to characterize new strains with new genes that can be used for this purpose. However, another important edge of the bacillus genus is its ability to cause food poisoning. In this sense, multiple studies characterize the strains with similar methodologies. They are currently published, for which we consider that we are in the current trends regarding the investigation of this microorganism. We present some examples in which, even over time, the characterization of the strains is similar. One of the main changes is the techniques to evaluate genetic diversity; however, in the discussion section, we specify that the method used obtains comparable results (lines 359- 363).*

*Sánchez-Chica, J., Correa, M. M., Aceves-Diez, A. E., & Castañeda-Sandoval, L. M. (2021). Genetic and toxigenic diversity of Bacillus cereus group isolated from powdered foods. Journal of Food Science and Technology, 58(5), 1892-1899. https://doi.org/10.1007/s13197-020-04700-2*

*Samapundo, S., Heyndrickx, M., Xhaferi, R., & Devlieghere, F. (2011). Incidence, diversity and toxin gene characteristics of Bacillus cereus group strains isolated from food products marketed in Belgium. International Journal of Food Microbiology, 150(1), 34-41. https://doi.org/10.1016/j.ijfoodmicro.2011.07.013*

*Gao, T., Ding, Y., Wu, Q., Wang, J., Zhang, J., Yu, S., Yu, P., Liu, C., Kong, L., Feng, Z., Chen, M., Wu, S., Zeng, H., & Wu, H. (2018). Prevalence, Virulence Genes, Antimicrobial Susceptibility, and Genetic Diversity of Bacillus cereus Isolated From Pasteurized Milk in China. Frontiers in Microbiology, 9, 533. https://doi.org/10.3389/fmicb.2018.00533*

*Radmehr, B., Zaferanloo, B., Tran, T., Beale, D. J., & Palombo, E. A. (2020). Prevalence and Characteristics of Bacillus cereus Group Isolated from Raw and Pasteurised Milk. Current Microbiology, 77(10), 3065-3075. https://doi.org/10.1007/s00284-020-02129-6*

*Rana, N., Panda, A. K., Pathak, N., Gupta, T., & Thakur, S. D. (2020). Bacillus cereus: Public health burden associated with ready-to-eat foods in Himachal Pradesh, India. Journal of Food Science and Technology, 57(6), 2293-2302. https://doi.org/10.1007/s13197-020-04267-y*

Validity of the findings

Thirdly, the results and discussion parts were short and dull, the data presentation is poor. Due to the lack of significance data to be discussed, I suggest the authors to perform more in-depth study.

*We improve these sections. In the results section, we add some descriptions that seem to be necessary (lines 216-219, 226-232, 245-248). In the discussion section, we added more information with comparisons that we had not done (lines 282-310, 349-367)*

Reviewer 2 (Anonymous)

Basic reporting

The manuscript written by Cástulo-Arcos et al. reports some interesting results. In this work, the authors have discovered that Bacillus cereus strains produce a low amount of biofilm and produce rates sporulation around 80%.

Experimental design

In general, this paper is clearly laid out, well planed and easy to read. The experiments are well designed and appropriate controls are presented.

Validity of the findings

The novelty and significance of the manuscript were not highlighted, please modify the paper more clearly.

*We consider that we failed to justify the project adequately. We add this information in the introduction sections (lines 90-101).*

Additional comments

Manuscript Title: Genetic diversity of enterotoxigenic Bacillus cereus strains in coriander in southwestern Mexico

Manuscript ID: #71248

The manuscript written by Cástulo-Arcos et al. reports some interesting results. In this work, the authors have discovered that Bacillus cereus strains produce a low amount of biofilm and produce rates sporulation around 80%.

Some specific suggestions or questions are listed below:

1.Abstract: Abstract should be written more precisely and explain novelty of this work.

*We improve the abstract*

2.Introduction: Introduction is easy to read but needs a little completed. Bacillus strains are well known for their metabolic capability and environmental versatility as well as for their ability to manage bacterial and fungal pathogens infecting crop plants. Authors should add more information into this section and cite the recent research into the field based on the recent literature (such as doi: 10.3390/plants11030457; doi: doi: 10.3390/microorganisms10020365; 10.3390/microorganisms9122511; doi: 10.1016/j.jwpe.2020.101712; doi: 10.3390/plants10112342; doi:10.3390/microorganisms8020223; doi: 10.1016/j.postharvbio.2013.10.004; doi: 10.1007/s00253-014-6164-y; doi: 10.1016/j.biortech.2012.01.106; doi: 10.1016/j.chemosphere.2022.133609; doi: 10.1021/jf404908j; doi: 10.3389/fbioe.2020.570307).

We add the information (lines 65- 79)

3.Introduction: Coriandrum sativum, Please check throughout the manuscript that abbreviations/acronyms are defined the first time they appear in each of three sections: the abstract; the main text; the first figure or table.

*We check it*

4.Introduction: In addition, this part should focus on the research progress related to the topic and emphasize the innovation of this research. However, the novelty and significance of the topic were not highlighted, please modify the introduction more clearly.

*We add this information in the introduction sections (lines 90-101).*

5.Material and Methods: please add relative references to support all the methods.

*We add it*

6.Results: This section is too simple. Please add all the results in details.

*We improve*

7.Results: Figure 3 is not clear. Please re-draw the figure and provide high quality one.

*We, by densitometry analysis, change the bands of the GTG technique for lines. the position of the lines facilitate the interpretation of the image. In the new image we added, we combined the dendrogram, the gel and the lines. This presentation is taken from other works. We add an example*

*Bhowmick, P. P., Srikumar, S., Devegowda, D., Shekar, M., Darshanee Ruwandeepika, H. A., & Karunasagar, I. (2012). Serotyping & molecular characterization for study of genetic diversity among seafood associated nontyphoidal Salmonella serovars. The Indian journal of medical research, 135(3), 371–381.*

8.Discussion: Authors should add more depth discussion into this section and cite the recent research into the field.

*we improve the discussion, and we return to the relevance of the work, the information that you asked us to consult (lines 282-310, 351-363)*

9.Conclusions: Authors can add and revise the Conclusions section for the better understanding of the topic and its future research.

*We add it (lines 396- 399)*

References: Many of the references have been superceded and more modern ones are required

*We had some complications with the references. Both reviewers agreed that the discussion lacked depth. Therefore, we added information, but the number of citations increased. In addition, we add all the bibliographical references that you kindly recommended to us.*

*As for the old references, only some methodological and essential information on the microorganism remains.*